# Gender-Specific Differences in Human Vertebral Bone Marrow Clot

**DOI:** 10.3390/ijms241411856

**Published:** 2023-07-24

**Authors:** Francesca Salamanna, Deyanira Contartese, Veronica Borsari, Stefania Pagani, Maria Sartori, Matilde Tschon, Cristiana Griffoni, Gianluca Giavaresi, Giuseppe Tedesco, Giovanni Barbanti Brodano, Alessandro Gasbarrini, Milena Fini

**Affiliations:** 1Complex Structure Surgical Sciences and Technologies, IRCCS Istituto Ortopedico Rizzoli, 40136 Bologna, Italy; francesca.salamanna@ior.it (F.S.); veronica.borsari@ior.it (V.B.); stefania.pagani@ior.it (S.P.); maria.sartori@ior.it (M.S.); matilde.tschon@ior.it (M.T.); gianluca.giavaresi@ior.it (G.G.); 2Spine Surgery Unit, IRCCS Istituto Ortopedico Rizzoli, 40136 Bologna, Italy; cristiana.griffoni@ior.it (C.G.); giuseppe.tedesco@ior.it (G.T.); giovanni.barbantibrodano@ior.it (G.B.B.); alessandro.gasbarrini@ior.it (A.G.); 3Scientific Direction, IRCCS Istituto Ortopedico Rizzoli, 40136 Bologna, Italy; milena.fini@ior.it

**Keywords:** clot, vertebral bone marrow, gender, spinal fusion, bone

## Abstract

Recently, our group described the application of vertebral bone marrow (vBMA) clot as a cell therapy strategy for spinal fusion. Its beneficial effects were confirmed in aging-associated processes, but the influence of gender is unknown. In this study, we compared the biological properties of vBMA clots and derived vertebral mesenchymal stem cells (MSCs) from female and male patients undergoing spinal fusion procedures and treated with vBMA clot. We analyzed the expression of growth factors (GFs) in vBMA clots and MSCs as well as morphology, viability, doubling time, markers expression, clonogenicity, differentiation ability, senescence factors, Klotho expression, and HOX and TALE gene profiles from female and male donors. Our findings indicate that vBMA clots and derived MSCs from males had higher expression of GFs and greater osteogenic and chondrogenic potential compared to female patients. Additionally, vBMA-clot-derived MSCs from female and male donors exhibited distinct levels of HOX and TALE gene expression. Specifically, HOXA1, HOXB8, HOXD9, HOXA11, and PBX1 genes were upregulated in MSCs derived from clotted vBMA from male donors. These results demonstrate that vBMA clots can be effectively used for spinal fusion procedures; however, gender-related differences should be taken into consideration when utilizing vBMA-clot-based studies to optimize the design and implementation of this cell therapy strategy in clinical trials.

## 1. Introduction

Spinal fusion (SF) is a common orthopedic procedure used to treat spinal diseases [1,2]. In addition to fixation systems, bone grafting is often required to enhance surgical outcomes [1,2].

### 1.1. Vertebral Bone Marrow Clot

Cell-based therapies such as whole vertebral bone marrow aspirate (vBMA), which is obtained easily through transpedicular aspiration during spinal procedures, and bone allografts have been developed as alternatives to autografts for SF [1,2,3,4,5,6,7,8,9,10,11]. However, their usage is currently limited due to the lack of an effective processing method, the absence of a specific texture, and the possibility of dispersion away from the implant site. Recently, our research group described a potential solution: the use of a powerful formulation known as the vBMA clot [12]. This clot is formed naturally from bone marrow and maintains all the elements of vBMA within a matrix molded by the clot. The vBMA clot’s beneficial effects are attributed to its three-dimensional (3D) matrix, which facilitates the delivery of mesenchymal stem cells (MSCs) and alpha granules, platelet-specific proteins, cytokines/chemokines, growth factors, coagulation factors, and adhesion molecules [12,13]. To validate the clinical use of vBMA clots for spinal bone regeneration, a pilot clinical study is currently underway at our institute (CE-AVEC 587/2020/Sper/IORS). Despite the fact that intrinsic factors such as donor age or the presence of metabolic bone diseases may cause differences in the biological qualities of MSCs within BMA, a recent study demonstrated that the vBMA clot can be easily used without compromising cell viability, proliferation, and differentiation, even in aged patients [14]. Understanding the innate properties, proliferative abilities, and differentiation potential is crucial for the potential future use of vBMA clots in the context of personalized tissue engineering and regenerative medicine.

### 1.2. Gender Differences

Another crucial aspect to consider is the influence of gender on the behavior and response of vBMA-clot cells. Gender differences have been studied extensively in various musculoskeletal diseases [15,16,17,18]. While past research mainly attributed these differences to the regulatory effects of gonadal hormones, the underlying causes are still largely unknown, with one possible explanation being gender-based variations in cells responses to the local microenvironment [18,19,20]. Gender-based differences in cell number, proliferative ability, differentiation, and trophic factor production have been observed in iliac crest bone-marrow-derived mesenchymal stem cells (BMSCs) across different species [15,16]. Female mice and rats, for instance, have been found to have fewer BMSCs in their bone marrow of compared to males [15,16]. It is suggested that this difference in progenitor cell number contributes to more effective healing of femoral bone defects in male rats compared to female rats [16]. Katsara et al. observed that male mouse BMSCs exhibited greater osteogenic and adipogenic potential than female cells, although gender did not affect proliferation [15]. Similarly, human studies have demonstrated a negative correlation between female donors and osteogenesis as well as collagen type I production [21,22,23,24]. Therefore, as emphasized by these studies and by National Institutes of Health (NIH), the role of stem cell gender must be carefully considered and evaluated in the field of tissue engineering and regenerative medicine. However, to date, no studies have assessed gender differences in MSCs derived from clotted vBMA.

In the present study, we compared the biological and molecular properties of human male and female MSCs derived from vBMA clots. We evaluated MSCs phenotyping, population-doubling time, colony-forming units, in vitro differentiation ability, growth factors secretion, senescence-associated factor expression, and HOX and TALE gene expression to characterize the gender-based efficacy of vBMA-clot cell therapy.

## 2. Results

### 2.1. Enzyme-Linked Immunosorbent Assay (ELISA)

Supernatant from female and male BMAs clots dissolved after 72 h of culture did not show significant difference for bFGF (*p* < 0.510), VEGF (*p* < 0.977), PDGF-AB (*p* < 0.220), and PDGF-C (*p* < 0.153) growth factors synthesis. Instead, significant, higher values of BMP-2 and TGF-β1 were detected in BMAs clots from male vs. female donors (*p* < 0.0005) (Figure 1). TGF-β1 has an anabolic and anti-inflammatory effect on MSCs proliferation and colonies formation, while BMP2 is a potent osteo-inductive molecule able to induce osteogenic differentiation of responsive cells.

### 2.2. Cell Morphology and Viability

LIVE/DEAD staining proved the viability of MSCs from females’ and males’ clotted vBMAs. At day 15, the average number of live cells for MSCs from females’ clotted vBMAs was found to be 95,000  ±  125 cells/mL, with an average number of dead cells of 100  ±  15 cells/mL, while for MSCs from males’ clotted vBMAs, the average number of live cells was 100,000  ±  205 cells/mL, with an average number of dead cells of 150  ±  15 cells/mL (Figure 2). No significant differences were present between MSCs from females’ and males’ clotted vBMAs.

### 2.3. Flow Cytometry

Flow cytometry was conducted to confirm MSCs phenotype. We observed that MSCs from females’ and males’ clotted vBMAs maintained 96% or greater expression of CD73, CD90, and CD105 and lacked expression of hematopoietic stem cell marker CD34, hematopoietic lineage marker CD45, and endothelial cells marker CD31 (Figure 3).

### 2.4. Population-Doubling Time (PDT)

No significant differences were observed between MSCs from females’ and males’ clotted vBMAs for PDT (*p* < 0.325) (Figure 4).

### 2.5. Colony-Forming Units (CFUs) Assay

No significant differences were detected between MSCs from females’ and males’ clotted vBMAs for CFUs (*p* < 0.598) (Figure 5).

### 2.6. Osteogenic, Adipogenic, and Chondrogenic Differentiation Ability

Osteogenic-induced MSCs from females’ and males’ clotted vBMAs showed presence of calcium deposits. However, in MSCs from males’ clotted vBMAs, a higher amount of mineralized matrix was seen (Figure 6A). These data were further confirmed by RT-PCR gene expression. Here, we evaluated RUNX2, the master regulator of osteogenic differentiation that in turn controls the expression of COL1A1, ALP, and BGLAP/osteocalcin. Out of all genes regulated by RUNX2, ALP, a cell surface protein ubiquitously expressed by several cell types, is used as marker for screening pre-osteoblasts. COL1A1 is an early osteoblast marker, while BGLAP is a late differentiation marker expressed in the bone, with a specific function associated with mineralization and matrix synthesis. Finally, OPG is expressed by osteoblasts and MSCs and can enhance osteogenesis by acting as a decoy receptor for RANKL, inhibiting osteoclastogenesis. Our results showed that MSCs from males’ clotted vBMA had significantly higher expression of COL1A1 and OPG (COL1A1, OPG: *p* < 0.0005) in comparison to MSCs from females’ clotted vBMA (Figure 6A).

A reduction in cell density and a change of cells shape were detected in all culture conditions when MSCs from females’ and males’ clotted vBMAs were cultured with adipogenic medium (Figure 6B). RT-PCR results indicated that both cultures, female and male, were able to differentiate onto the adipogenic lineage (ADIPO Q, PPARg) when exposed to definite inducing factors, without any significant differences (Figure 6B).

Concerning the chondrogenic differentiation of MSCs derived from females’ and males’ clotted vBMAs, the presence of chondrocytes and extracellular matrix was detected in all culture conditions (Figure 6C). RT-PCR revealed a significant, higher expression of SOX9 (*p* < 0.0005), a chondrogenic gene that promotes progression to chondrocyte differentiation at later stages, in MSCs derived from males’ clotted vBMA in comparison to females’ BMAs (Figure 6C).

### 2.7. Gene Expression of HOX and TALE Genes

After osteogenic induction HOXA1 (*p* < 0.05), HOXB9 and HOXA11 (*p* < 0.005) and HOXB8 and PBX1 (*p* < 0.0005) showed significantly higher values in MSCs from males’ clotted vBMA in comparison to females’ vBMA (Figure 7).

### 2.8. Senescence Associated Factors Expression

Klotho and senescence-associated genes (IL1β, IL1α, IL6, IL8, CCL4, CXCL2, TNFα, and MCP-1) expression showed no significant difference between vBMA from female and male donors (Figure 8).

## 3. Discussion

Human MSCs derived from different tissues have demonstrated gender-related differences in proliferative abilities, differentiation potential, production of trophic factors, dynamics clone development, and surface antigen expression [25]. Initially, these changes were primarily attributed to the differential regulatory effects of gonadal hormones [26]. However, evidence suggests that gender differences may not solely correlate with modifications in circulating sex steroid levels but also with intrinsic changes in target cells [27]. Therefore, understanding the innate properties and potential use of BMA and MSCs derived from various anatomical sites and different formulations (whole, purified, expanded, or concentrated) in regenerative medicine strategies while considering gender-related influences is critically important for improving therapeutic efficacy and minimizing adverse events. In this study, we evaluated the gender-related differences of a new formulation of vBMA, namely the vBMA clot, derived from female and male patients undergoing SF surgery and treated with this new formulation of vBMA.

### Gender-Related Differences in vBMA Clot

In this study, we observed that vBMA clots and derived MSCs are influenced by gender-related differences primarily associated with osteoblasts maturation and differentiation. We found specific differences in the expression of TGF-β1 and BMP-2 proteins in the supernatant of dissolved females’ and males’ vBMAs clots after 72 h of culture with higher levels detected in male donors. The formation of vBMA clots involves platelets degranulation [28], which delivers osteotropic cytokines and growth factors, including TGF-β and BMP-2 [28,29]. TGF-β1, the most abundant isoform of the TGF-β superfamily, it primarily sourced from platelets (20 mg/kg) and, along with BMP-2, plays a central role in regulating cell proliferation, differentiation, MSCs osteogenic induction, and extracellular matrix (ECM) remodeling [30,31,32,33]. RT-PCR analyses further confirmed these data, showing a negative correlation in osteogenesis and production of COL1A1 and OPG in MSCs derived from females’ vBMA clots. These results are consistent with previous literature data showing that male bone marrow MSCs have stronger osteogenic potential than female cells, although our study on vertebral MSCs did not find any effects of gender on morphology, viability, surface marker expression, colony-forming units, and proliferative abilities [15,34,35]. Here, we also demonstrated higher expression levels of specific growth factors involved in bone regeneration and healing as well as defined osteogenic markers (TGF-β1, BMP-2, COL1A1, and OPG) in vBMA clots derived from male donors, indicating superior bone formation ability.

Furthermore, we observed that the chondrogenic differentiation ability, as verified by SOX9 expression, was higher in MSCs derived from male vBMA clots. SOX9, a regulator of chondrocyte phenotype, is a key gene in chondrogenic differentiation and extracellular matrix production [36].

Contrasting osteogenic and chondrogenic differentiation ability, no significant differences were found in adipogenic differentiation potential between MSCs derived from males’ and females’ vBMA clots, suggesting that gender does not affect adipogenic differentiation potential. Similar results were obtained in this study for the expression of senescence-associated factors, which can significantly impact the clinical therapeutic potential of MSCs as well as their paracrine effects, immunomodulatory activity, and cell migration and ability.

Since our previously studies on HOX and TALE genes expression showed that bone marrow MSCs derived from different body sites (iliac crest, sternum, and vertebrae), displayed distinct HOX and TALE signatures [37], we evaluated whether MSCs isolated from males’ and females’ vBMA clots might, following osteogenic induction, modulate HOX and/or TALE expression. Our results highlighted that MSCs isolated from males’ and females’ vBMA clots can be identified for HOX and TALE expression levels. Specifically, MSCs derived from males’ vBMA clots upregulated HOXA1, HOXB8, HOXD9, HOXA11, and PBX1. HOXA1, for instance, has been identified as a key factor involved in osteogenesis of human MSCs, with bone sialoprotein and COL1A1 being target molecules of HOXA1 [38]. HOXA11, another HOX gene within the A family that showed a critical role in osteogenesis, was upregulated in MSCs derived from males’ vBMA clots. Increases in HOXA11 expression at the injury site have been observed after fractures, indicating its importance in fracture healing, chondrocytes maturation, and extracellular matrix remodeling [39,40]. Additionally, HOXA11 is required for physiological bone turnover, as it controls osteocyte renewal, stimulates osteoblast maturation, and maintains structural integrity of collagen fibrils in bone [41]. HOXB8, upregulated in MSCs derived from males’ vBMA clots, is involved in the expansion of hematopoietic stem and early progenitor cells and vertebral development. HOXB8 deactivation can lead to slight vertebral abnormalities, while its increase may can help overcome the blockade of posterior elongation of axial tissues [37]. HOXB8 may also play a significant role in the regulation of adult MSCs, serving as a marker of vertebral identity and as regulator of MSCs expansion. Within the D family of the HOX cluster, which has been associated with MSCs condensation and further development into chondrocytes, we observed upregulation of HOXD9 in MSCs derived from males’ vBMA clots. HOXD9 is distinctly expressed in developing chondrocytes within MSCs [42]. Finally, PBX1, a TALE class factor that controls several embryonic processes [43,44,45,46,47,48], was upregulated in MSCs derived from males’ vBMA clots. PBX1 has been implicated in supporting progenitor cell proliferation in numerous tissues [49], and while its role in adult tissue stem cells is undetermined, it is prevalently expressed in long-term hematopoietic stem cells [50,51].

These results suggest that vBMA clots can be effectively used for SF procedures, but altered osteogenic differentiation ability and specific levels of HOX and TALE signatures are present in female patients. To date, the reason for these intrinsic differences between female and male MSCs are poorly understood and directly linked to specific responses of cells to the microenvironment [52]. For example, it is known that fracture risk is higher in females compared to males [53]; females also showed a higher incidence of stress fracture early in life [54]. In addition, a clinical study showed that female gender was linked to a decreased fracture union rate [55] and was recognized as a major risk factor for compromised bone healing in various clinical studies [56,57]. However, despite all this information, to date, preclinical and clinical research still seldom consider the gender-related differences in regenerative medicine and tissue engineering strategies. Although further studies are mandatory, the knowledge of the appropriate expression of HOX and TALE genes associated with a detailed characterization of the vBMA clot and its derived MSCs is of key importance for a preliminary clarification of the differences within and between females and males and for the improvement of the basic understanding of stem cell biology and future design and use of this cell therapies strategy in clinical trials.

## 4. Materials and Methods

### 4.1. Isolation and Culture of Human BMA Clot

In a pilot clinical study on the efficacy of clotted vBMA for SF surgery (Ethical Committee approval: CE-AVEC 587/2020/Sper/IORS), we demonstrated the efficacy of this new BMA formulation. Because the focus of these studies is bone regeneration in patients submitted to SF surgery, it is of relevant interest to include all biological and molecular patterns that could influence the results, such as gender-related differences in vBMA clots and in the derived MSCs. Inclusion and exclusion criteria for this pilot clinical study were detailed previously [14]. Written informed consent was obtained from all patients that participated in the pilot clinical study. Briefly, human BMA was collected from the vertebral pedicles of 6 patients, 3 females (mean age: 44 ± 0.5) and 3 males (mean age: 41 ± 4.6), undergoing to spinal surgical procedures. As previously reported, 1 or 2 mL of vBMA from each vertebra was harvested and put in a sterile container with no anticoagulant. vBMA clotted in ~15–20 min and was placed in culture flasks with Dulbecco’s modified Eagles (DMEM) complete medium (DMEM, 10% fetal bovine serum, 100 U/mL penicillin, 100 μg/mL streptomycin, and 5 μg/mL plasmocin) (Sigma-Aldrich, St. Louis, MO, USA; Lonza; Gibco Life Technologies, Carlsbad, CA, USA; Invivogen, San Diego, CA, USA). The culture flasks were incubated at 37 °C in 5% CO_2_ and under hypoxic condition (2% O_2_) and medium changed when 80–90% of cells were adherent.

### 4.2. Enzyme-Linked Immunosorbent Assay (ELISA)

Once the clot was dissolved (after 72–96 h of culture), media were collected and centrifuged to eliminate particulates, and aliquots of supernatant were stored at −20 °C. Immuno-enzymatic assays, an enzyme-linked immunosorbent assay (ELISA), were employed to quantify the amount of basic fibroblast growth factor (bFGF) (pg/mL), vascular endothelial growth factor (VEGF) (pg/mL), transforming growth factor beta 1 (TGF-β 1) (pg/mL), bone morphogenetic protein 2 (BMP-2) (pg/mL), platelet-derived growth factor AB (PDGF-AB) (pg/mL), and platelet-derived growth factor C (PDGF-C) (pg/mL) (Sigma–Aldrich, St. Louis, MO, USA) according to the manufacturer’s instructions. The absorbance was measured at 450 using an ELISA reader (Imark Microplate Reader, ELISA-Biorad SRL). Each sample was tested in triplicate.

### 4.3. Cell Morphology and Viability

MSCs from females’ and males’ clotted vBMAs were observed twice a week with a light microscope (Nikon Eclipse, Milan, Italy). At confluence, after about 14 days, the viability of cells was evaluated by LIVE/DEAD^®^ assay (Thermo Fisher Scientific, Waltham, MA, USA) according to the manufacturer’s instructions. Briefly [14], MSCs were incubated with calcein-AM (4 μM) and ethidium homodimer-1 (2 μM) for 45 min at 37 °C. Images were acquired by a fluorescence microscope (Nikon Eclipse, Italy) equipped with a digital camera. The green and red fluorescence was characteristic of live and dead cells, respectively. The number of dead cells per cm^2^ was determined. Each sample was tested in triplicate.

### 4.4. Flow Cytometry

Antigen expression was performed as previously described [14] and assessed with FACSCanto II instrument (Becton Dickinson, Franklin Lakes, NJ, USA) and by FACSDiva software 6.0 (Becton Dickinson). Briefly, at passage 1, 0.5–1 × 10^5^ of MSCs for each antigen were washed with PBS, centrifuged, and incubated in flow cytometry buffer, adding fluorescein isothiocyanate (FITC)-conjugated antibody against CD-31, 45, 34, 44, 73, 90, and 105. FITC-conjugated nonspecific immunoglobulin G (IgG) was used as control (Bio-Legend, San Diego, CA, USA).

### 4.5. Population-Doubling Time

Population-doubling time (PDT) was assessed as previously described [14] at passage 1. Cells were plated at 7 × 10^3^ cells/cm^2^ and, after 10 days, evaluated by erythrosine vital dye. Cumulative population doublings (CPDs) and PDT were determined as reported below:CPDs = log (N/N0) × 3.31
PDT = CT/CPDs

N: final number of cells; N0: initial number of cells; CT: time in culture. Each sample was tested in triplicate.

### 4.6. Colony-Forming Assay

At passage 1, 200 MSCs/cm^2^ were plated and cultured for 10 days to assess the number of colony-forming units (CFUs). Cells were fixed in 10% formalin and stained with toluidine blue [14]. Aggregates with ≥20 cells were scored as colonies and counted (Olympus BX51). Each sample was tested in triplicate.

### 4.7. Osteogenic, Adipogenic, and Chondrogenic Differentiation Ability

As previously described [14], for osteogenic or adipogenic differentiation, MSCs were plated at a density of 7 × 10^3^ cells per cm^2^ and incubated in DMEM complete medium. After 24 h, osteogenic (DMEM complete medium, dexamethasone 10^−8^ M, ascorbic acid 50 μg/mL, and β-glycerophosphate 10 mM) or adipogenic (DMEM complete medium, isobutylmethylxanthine 500 μM, indomethacin 100 μM, dexamethasone 1 × 10^−6^, and insulin 10 μg/mL) medium was added, and MSCs were cultured for 15 days. Chondrogenesis was induced by pelleting 2.5 × 10^5^ cells at 260 g for 20 min (micromasses). After 24 h, chondrogenic medium (5 μg/mL insulin, 5 μg/mL transferrin, 5 μg/mL selenous acid, 0.1 μM dexamethasone, 0.17 mM ascorbic acid–2-phosphate, 1 mM sodium pyruvate, 0.35 mM proline, and 10 μg/mL transforming growth factor-β3 (TGF-β3)) was added, and pellets were cultured for 30 days.

### 4.8. Staining

Osteogenic and adipogenic cultures were fixed, respectively, in 10% formaldehyde and 4% PFA and subsequently stained with Alizarin Red S (Sigma-Aldrich) and 1.8% Oil Red O (Sigma-Aldrich), respectively. LIVE/DEAD staining was applied to confirm cell viability. Chondrogenic micromasses were fixed in 10% formaldehyde and embedded in paraffin (Thermo Fisher Scientific, Waltham, MA, USA). Paraffin blocks were sectioned, and 3 sections for each sample were stained with Alcian Blue–Nuclear Fast Red and subsequently evaluated by a digital pathology scanner (Aperio-ScanScope, Leica Biosystems, Wetzlar, Germany). Each sample was tested in triplicate.

### 4.9. Gene Expression

Total RNA was extracted MSCs that differentiated toward osteogenic, adipogenic, and chondrogenic lineages, as previously reported [14]. RNA extraction was carried out using a RNeasy Mini Kit (Ambion by Life Technologies, Carlsbad, CA, USA) and quantified by a spectrophotometer (NANODROP 2720, Applied Biosystem, Waltham, MA, USA), reverse transcribed using the Superscript Vilo cDNA synthesis kit (Life Technologies, Carlsbad, CA, USA), and diluted to a concentration of 5 ng/μL. Then, 10 ng of cDNA for each sample was tested in duplicate. Gene expression was evaluated by semiquantitative PCR analysis using the SYBR green PCR master mix (QIAGEN GmbH, Hilden, Germany) in a LightCycler 2.0 Instrument (Roche Diagnostics, Basel, Switzerland; GmbH, Hilden, Germany; Manheim, Atlanta, GA, USA). The protocol comprised a denaturation cycle at 95 °C for 15 min, 25 to 40 cycles of amplification, and a melting curve analysis to check for amplicon specificity. Primers’ details are reported in Table 1. The mean threshold cycle was used for the calculation of relative expression using the Livak method (2^−ΔCt^), with GAPDH as the reference gene [58]. Each sample was tested in triplicate.

### 4.10. Gene Expression of HOX and TALE Genes

The cDNA of osteogenic cells was also investigated for HOX and TALE genes expression, as previously described. Primers are listed in Table 1. Each sample was tested in triplicate.

### 4.11. Senescence Associated Factors Expression

As previously described [14], at passage 1, 7 × 10^3^ MSCs/cm^2^ were cultured with DMEM complete medium for 10 days to evaluate senescence-associated gene expression. Primers details for all genes are described in Table 1. Each sample was tested in triplicate.

### 4.12. Statistical Analysis

Statistical analysis was carried out by using the IBM^®^ SPSS^®^ Statistics v.23 software (Armonk, NY, USA). Data are reported as mean ± standard deviation (SD) at a significant level of *p* < 0.05. After verification of normal distribution and homogeneity of variance (Levene test), data were assessed with *t*-test to compare females’ and males’ MSCs from vBMAs clot.

## Figures and Tables

**Figure 1 ijms-24-11856-f001:**
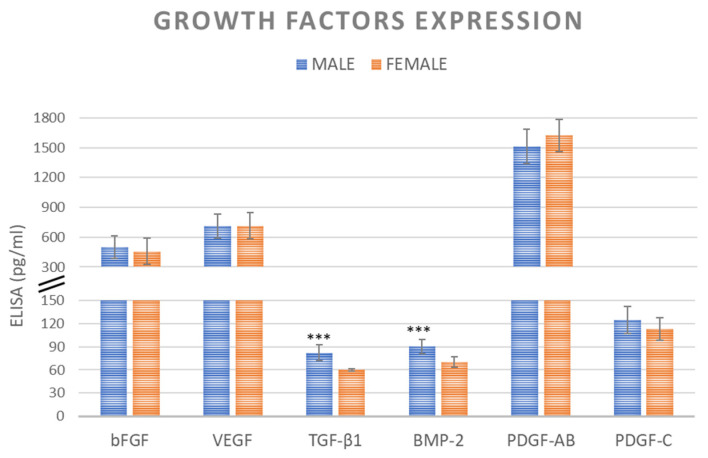
bFGF (pg/mL), VEGF (pg/mL), TGF-β 1 (pg/mL), BMP-2 (pg/mL), PDGF-AB (pg/mL), and PDGF-C (pg/mL) ELISA expression from female and male vBMAs clots dissolved after 72 h of culture. The absorbance was measured at 450 nm. The error bars represent the standard deviation between the samples. BMP-2 and TGF-β1: *** male vs. female (*** *p* < 0.0005).

**Figure 2 ijms-24-11856-f002:**
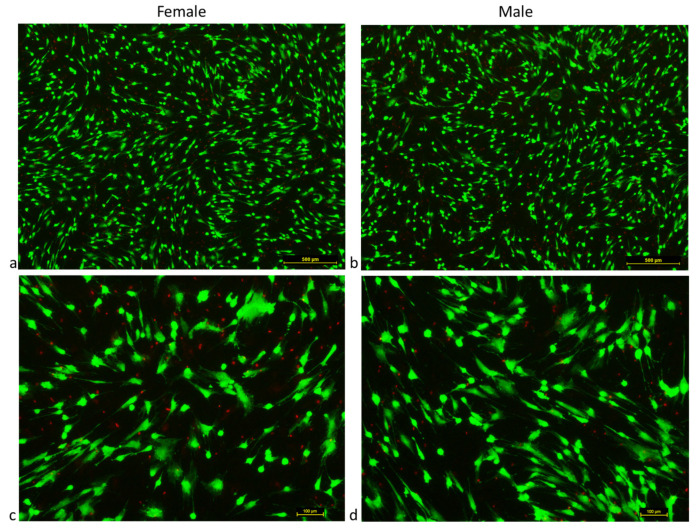
LIVE/DEAD endogenous cell staining of MSCs from females’ and males’ clotted vBMAs after 15 days of culture. Live cells stained in green with calcein-AM. Dead cells stained red with ethidium homodimer-1; (**a**,**b**) magnification 4×; (**c**,**d**) magnification 20×. Scale bar: (**a**,**b**) 500 µm; (**c**,**d**) 100 µm.

**Figure 3 ijms-24-11856-f003:**
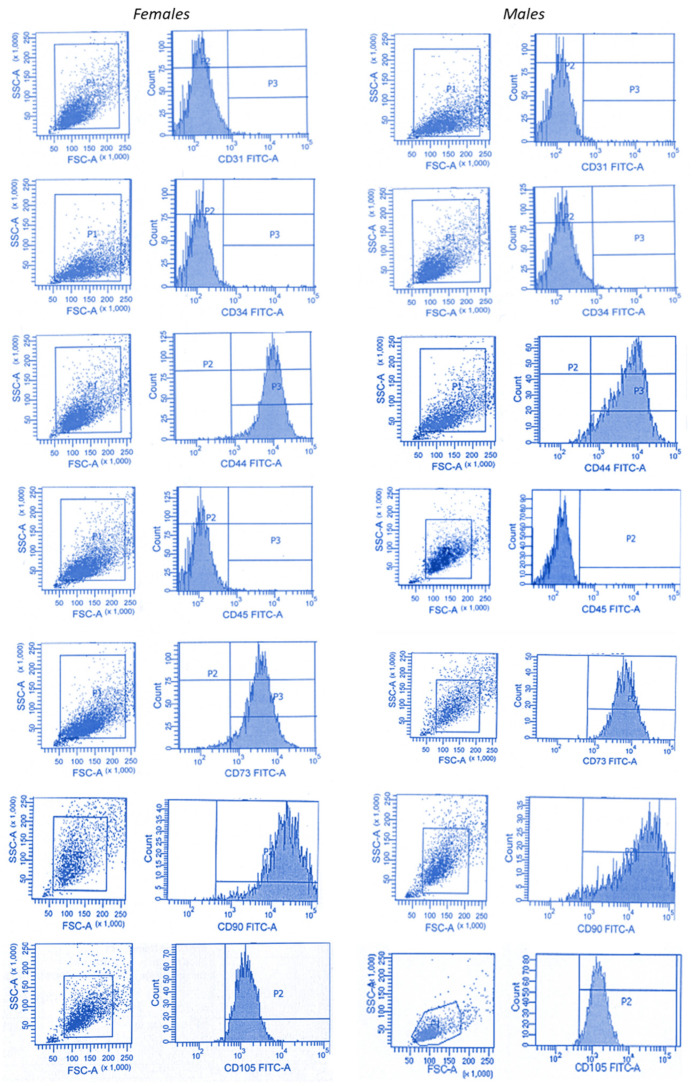
Representative flow cytometry graph of MSCs from females’ and males’ clotted vBMAs, using fluorescent activated cell sorting (FACS). The MSCs-positive surface CD markers, namely CD44, CD73, CD90, and CD105, and the MSCs-negative surface CD markers, namely CD31, CD34, and CD45, are shown. The independent CD marker antigens were tagged with different fluorochromes. All the MSC-positive CD surface markers demonstrated more than 90% positivity, and their histograms were considerably shifted to the right compared to their respective isotype controls.

**Figure 4 ijms-24-11856-f004:**
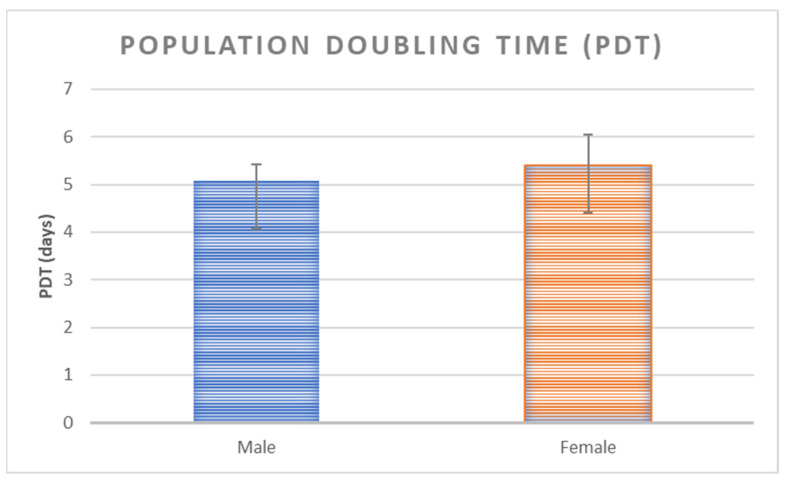
PDT for MSCs from vBMAs clots from female and male donors.

**Figure 5 ijms-24-11856-f005:**
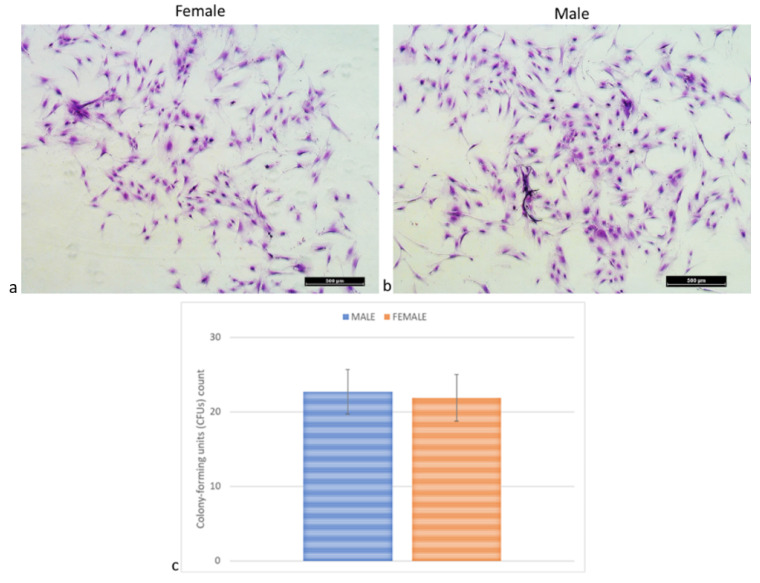
CFUs count of MSCs from vBMAs clots from female and male donors after 10 days of culture in basal medium. (**a**,**b**) Microscopic images of a single stained colony. Magnification 10×. (**c**) Number of positive colonies/well by toluidine blue staining. Scale bar: (**a**,**b**) 500 µm.

**Figure 6 ijms-24-11856-f006:**
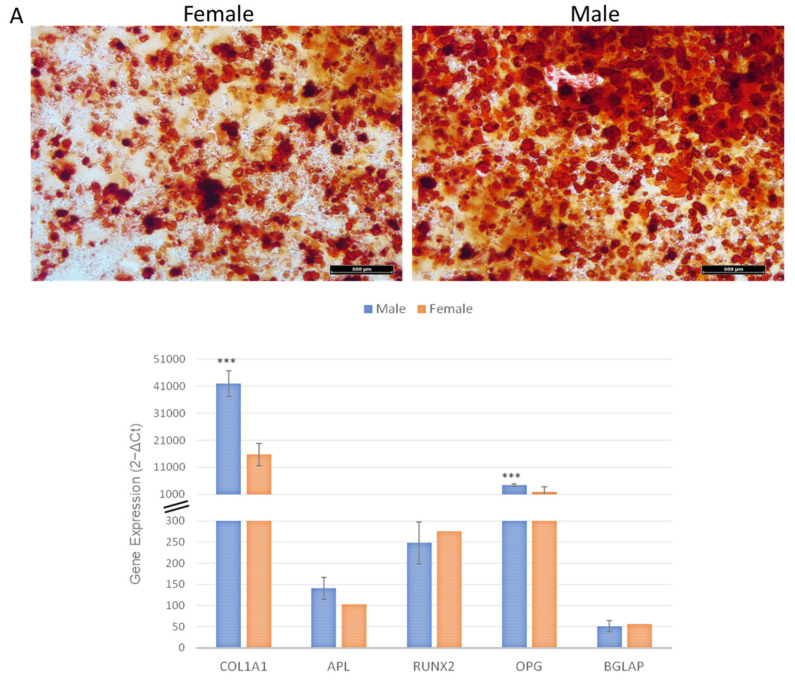
Representative images and gene expression profile of (**A**) osteogenic (Alizarin Red S staining, magnification 4×, scale bar: 500 µm), (**B**) adipogenic (Oil Red O, magnification 40×, scale bar: 50 µm), and (**C**) chondrogenic (Alcian Blue/Nuclear Fast Red staining, magnification 80×, scale bar: 10 µm) differentiation of MSCs from females’ and males’ clotted vBMAs. Graphs represent gene expression profile evaluated through RT-PCR. The mean threshold cycle was used for the calculation of relative expression (2^−ΔCt^), with GAPDH as the reference gene. The error bars represent the standard deviation between the samples. *** male vBMA vs. female vBMA. *** *p* < 0.0005. Each sample was tested in triplicate.

**Figure 7 ijms-24-11856-f007:**
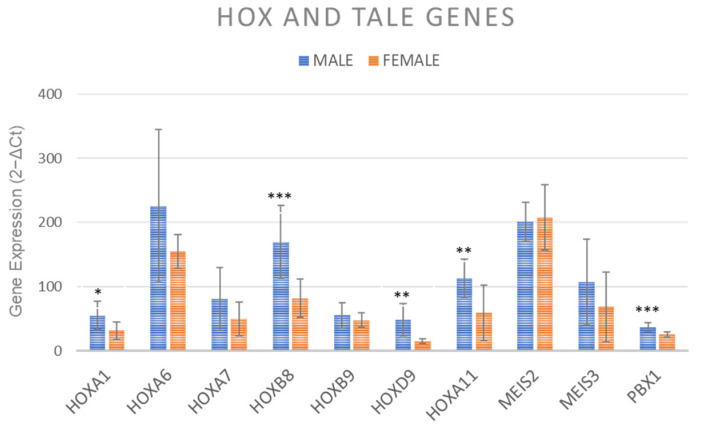
Gene expression measured by RT-PCR of HOX and TALE signatures for MSCs derived from females’ and males’ clotted vBMAs after 15 days of osteogenic induction. The mean threshold cycle was used for the calculation of relative expression (2^−ΔCt^), with GAPDH as the reference gene. The error bars represent the standard deviation between the samples. HOXA1, * male vs. female; HOXD9 and HOXA11, ** male vs. female; HOXB8 and PBX1, *** male vs. female. * *p* < 0.05; ** *p* < 0.005; *** *p* < 0.0005. Each sample was tested in triplicate.

**Figure 8 ijms-24-11856-f008:**
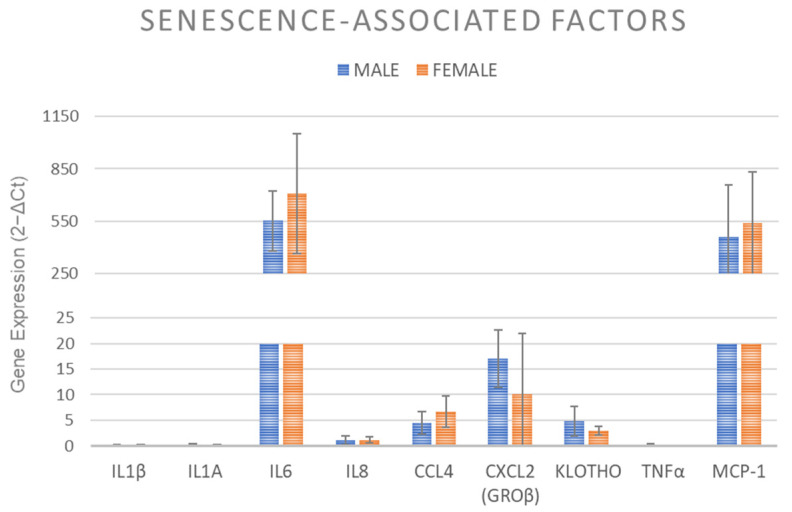
Gene expression measured by RT-PCR of MSCs derived from clotted vBMAs from female and male donors for Klotho and senescence-associated genes, i.e., IL1β, IL1α, IL6, IL8, CCL4, CXCL2, TNFα, and MCP-1. The mean threshold cycle was used for the calculation of relative expression (2^−ΔCt^), with GAPDH as the reference gene. The error bars represent the standard deviation between the samples. Each sample was tested in triplicate.

**Table 1 ijms-24-11856-t001:** Primers for qPCR analysis of gene expression.

GENE	Primer Forward	Primer Reverse	Annealing Temperature
GAPDH	5′-TGGTATCGTGGAAGGACTCA-3′	5′-GCAGGGATGATGTTCTGGA-3′	56 °C
ACAN	5′-TCGAGGACAGCGAGGCC-3′	5′-TCGAGGGTGTAGCGTGTAGAGA-3′	60 °C
SOX 9	5′-GAGCAGACGCACATCTC-3′	5′-CCTGGGATTGCCCCGA-3′	60 °C
COL2A1	QuantiTect Primer Assay (Qiagen) Hs_COL2A1_1_SG		55 °C
COL1A1	QuantiTect Primer Assay (Qiagen) Hs_COL1A1_1_SG		55 °C
ALPL	QuantiTect Primer Assay (Qiagen) Hs_ALPL_1_SG		55 °C
BGLAP	QuantiTect Primer Assay (Qiagen) Hs_BGLAP_1_SG		55 °C
RUNX2	QuantiTect Primer Assay (Qiagen) Hs_RUNX2_1_SG		55 °C
PPARg	QuantiTect Primer Assay (Qiagen) Hs_PPARG_1_SG		55 °C
ADIPOQ	QuantiTect Primer Assay (Qiagen) Hs_ADIPOQ_1_SG		55 °C
IL1A	QuantiTect Primer Assay (Qiagen) Hs_IL1A_1_SG		55 °C
IL 1B	QuantiTect Primer Assay (Qiagen) Hs_IL1B_1_SG		55 °C
IL 6	QuantiTect Primer Assay (Qiagen) Hs_IL6_1_SG		55 °C
CCL4L1 (MIP-1β)	QuantiTect Primer Assay (Qiagen) Hs_CCL4L1_4_SG		55 °C
CXCL2 (MIP-2α)	QuantiTect Primer Assay (Qiagen) Hs_CXCL2_1_SG		55 °C
KL (Klotho)	QuantiTect Primer Assay (Qiagen) Hs_KL_1_SG		55 °C
TNFα	QuantiTect Primer Assay (Qiagen) Hs_TNF_1_SG		55 °C
MCP-1	QuantiTect Primer Assay (Qiagen) Hs_CCL2_1_SG		55 °C
IL8	5′-ATGACTTCCAAGCTGGCCGTG-3′	5′-TTATGAATTCTCAGCCCTCTTCAAAAACTTCTC-3′	60 °C
HOXA1	Hs_HOXA1_1_SG		55 °C 20″
HOXA6	Hs_HOXA6_1_SG		55 °C 20″
HOXA7	Hs_HOXA7_2_SG		55 °C 20″
HOXA11	Hs_HOXA11_2_SG		55 °C 20″
HOXB8	Hs_HOXB8_2_SG		55 °C 20″
HOXB9	Hs_HOXB9_1_SG		55 °C 20″
HOXD9	Hs_HOXD9_1_SG		55 °C 20″
HOXD10	Hs_HOXD10_1_SG		55 °C 20″
MEIS2	Hs_MEIS2_1_SG		55 °C 20″
MEIS3	Hs_MEIS3_1_SG		55 °C 20″
PBX1	Hs_PBX1_1_SG		55 °C 20″

## Data Availability

The data presented in this study are available on request from the corresponding author.

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
