# Peer review of "Gender-Specific Differences in Human Vertebral Bone Marrow Clot"

_ijms, 2023, doi:10.3390/ijms241411856_

Round 1
Reviewer 1 Report
In this manuscript, Salamanna and coauthors conducted various experiments to compare vertebral bone marrow aspirate (vBMA) clot and derived vertebral mesenchymal stem cells (MCSs) from female and male donors, and analyzed their biological properties such as vBMA clot growth factor expression and MSC viability, markers expression, etc. The authors identified some obvious biological differences in both vBMA clot and MSCs from female and male donors, warning existence of gender-related differences in vBMA clot-based studies and clinical trials. Overall, the manuscript is logically clear, well written and easy to understand. The research method and statistical analyses are appropriate to address the question. However, there are many issues that should be addressed before considering for publication.
1. Both the Introduction and Discussion in the manuscript are organized into one paragraph with lots of information, which makes it very difficult to read. It is recommended to reorganize the two sections into logically clear and shorter paragraphs.
2. In Fig 1, the y-axis title (protein expression) is too vague to help interpret the values beside it. Also, the authors did not provide necessary details in both the Figure legend (lines 85-86) and Materials and Methods (lines 302-308) about the detection method used to analyze the levels of these growth factors. I assume that the authors used ELISA for the analysis; if so, the experimental details about ELISA should be provided. The same issue applies to Figs 5 and 6.
3. For Fig 2, the authors mentioned in lines 315-316 in the Materials and Methods that the green and red fluorescences indicate live and dead cells, respectively. The authors showed counts of dead cells in lines 91 and 93. However, in the Fig 2 there were no cells with red fluorescence present. In addition, the number of MSCs in Fig 2a is obviously larger than that in Fig 2b, which is not consistent with the authors’ claim that male clotted vBMAs have more MSCs than the female counterpart. What’s the authors’ response?
4. For the flow cytometry result shown in Table 1, the authors should provide the original plots instead of a table as the plots are more informative and appropriate.
5. In Fig 3, the y-axis is not consistent with the PDT. The authors suppose to use time instead of the MSC counts for the y-axis.
6. For Fig 4, it will be more helpful if the authors can provide pictures showing presence of multiple colonies within the same field.
7. In line 158, “COL1A1: ***male vBMA vs. female vBMA; OPG: ***female vBMA vs. male vBMA” in which “male vBMA vs. female vBMA” and “female vBMA vs. male vBMA” should be consistent to avoid misinterpretation.
8. In line 168, “HOXB9 and HOXA11” should be “HOXD9 and HOXA11”.
9. In all the figure legends, the authors should provide more details that help interpret the results. The current figure legends are excessively brief and lack of essential descriptions.
10. In most of the figures, the scale bars are shown in different styles and difficult to read.
11. In the Materials and Methods, there are many inaccurate expressions about the number of cells plated in different experiments. For example, 0.5-1 × 105 of MSCs in line 321, 7 × 103 cells in line 328, and 7 × 103 cells in line 341.
12. Line 330, CPDS should be CPDs to be consistent.
13. The “Table 1” in lines 372, 380 and 384 should be “Table 2”.
14. In the Results section, the authors directly presented the detection results for all of those different genes/proteins without explaining why choosing those genes/proteins for analyses in the first place, which usually make readers confused and unable to follow the logical flow. I noticed that the authors provided some introductions about some of these genes/proteins in the Discussion section. However, it is recommended that the authors provide the necessary introductions for these genes/proteins in the Results section.
Overall, the English language is acceptable for understanding the manuscript. However, there are many language errors and mistakes detected in the manuscript. It is recommended to ask a native speaker for a final proofreading.
Author Response
Attached are responses to comments/suggestions.

Reviewer 2 Report
The vBMA has been proven to be an effective cell therapy strategy for bone regeneration in spinal surgery. This study by Francesca Salamanna demonstrated the gender-differences in the biological properties of vBMA clot and vertebral mesenchymal stem cells (MSCs), including morphology, viability, proliferation, capacity of osteogenic, adipogenic, and chondrogenic differentiation, release of related cytokines, expressions of HOX and TALE gene profiles. Fantastic results were found in this document. There are major comments with regard to the manuscript.
1. In general, this work is a descriptive study aimed at elucidating the differences in the biological properties and capacities of vBMA clot and vertebral mesenchymal stem cells. The lack of data about the transplantation of different vBMA or MSCs into the fractured animals in vivo is a defect. Could authors accomplish this problem in the revision or following work?
2. The authors provide detailed procedure of isolation and culture of human BMA clot. But the procedure of MSCs isolation, culture and identification is obscure.
3. Only three samples of male and female group were subjected to the study, which seemed not to be enough for the accurate statistical test. Please increase the sample size and provide the scatted dot plots with bar overlay in each figure.
4. In Figure1 and Figure5, please add the unit of longitudinal axis in the graph.
5. In Table1, the proportions of MSCs with different phenotype were presented. Please provide the original flow chart of flow cytometry.
6. In Figure6, representative images of osteogenic, adipogenic, and chondrogenic differentiation ability were presented. Please add the Western Blot of remarkable hallmarks with regard to the pathways in the process of osteogenic, adipogenic, and chondrogenic differentiation.
7. In Figure 7, except for Klotho gene, the senescence-associated genes listed are associated with inflammation. Are there other specific genes related to senescence?
8. Except for the MSCs,many other cellular components co-exist in the marrow microenvironment. Whether the cross-talk between MSCs and other cells exists in the bone regeneration?
Author Response
Comments and Suggestions for Authors
The vBMA has been proven to be an effective cell therapy strategy for bone regeneration in spinal surgery. This study by Francesca Salamanna demonstrated the gender-differences in the biological properties of vBMA clot and vertebral mesenchymal stem cells (MSCs), including morphology, viability, proliferation, capacity of osteogenic, adipogenic, and chondrogenic differentiation, release of related cytokines, expressions of HOX and TALE gene profiles. Fantastic results were found in this document. There are major comments with regard to the manuscript.
- In general, this work is a descriptive study aimed at elucidating the differences in the biological properties and capacities of vBMA clot and vertebral mesenchymal stem cells. The lack of data about the transplantation of different vBMA or MSCs into the fractured animals in vivo is a defect. Could authors accomplish this problem in the revision or following work?
Since the completely biological nature of vBMA and the numerous in vitro studies carried out by our group, we decided to not use more complex preclinical model, as in vivo models, also to reduce the ethical burden following the 3R principles, the guiding principles aimed at replacing/reducing/refining (3R) animal use and their suffering for scientific purposes. However, considering the potential of our preclinical data we conducted and just completed a pilot clinical study on vBMA clot use in patients with degenerative spinal pathologies that undergoing spinal fusion (Ethics Committee approval CE AVEC 587/2020/Sper/IOR). The study registered in ClinicalTrials.gov (NCT05936047) and just submitted to a clinical journal revealed the efficiency and the safety profile of vBMA clot as advanced bio-scaffold able to achieve posterior lumbar fusion in the treatment of degenerative spine diseases, laying the groundwork for a larger randomized clinical study.
- The authors provide detailed procedure of isolation and culture of human BMA clot. But the procedure of MSCs isolation, culture and identification is obscure.
We thank the reviewer for the question. In our study MSCs from vBMA were isolated and cultured as primary culture and as previously described by other authors (Methods Mol Biol. 2012;816:3-18, Gastroenterol Hepatol Bed Bench. 2017 Summer; 10(3): 208–213) and as reported in the manuscript. Briefly, MSCs isolation was carried out from human bone marrow clot and cultured in defined medium (DMEM, 10% fetal bovine serum, 100 U/ml penicillin, 100 μg/ml streptomycin, 5 μg/ml plasmocin). Cultures were maintained at 37°C in 5% CO2 and under hypoxic condition (2% O2) and medium was exchanged when most cells (80-90%) were adherent. Subsequently, at confluence, for MSCs identification we used the standard identification protocol proposed by the International Society for Cellular Therapy (Dominici, M. et al. (2006) Cytotherapy 8:315) which foresees the expression of a specific panel of surface antigens (CD-31, -45, -34, -44, -73, -90 and -105), the plastic adherence, the evaluation of clonogenicity, and the multipotency, i.e. the MSCs ability to differentiated into osteogenic, adipogenic and chondrogenic lineage.
- Only three samples of male and female group were subjected to the study, which seemed not to be enough for the accurate statistical test. Please increase the sample size and provide the scatted dot plots with bar overlay in each figure.
The sample size of our study was chosen to guarantee a high probability (statistical power) of highlighting a statistically significant difference (avoiding the type I error), provided that a certain difference δ actually exists. This analysis was based on preclinical data previously obtained using the vBMA clot (Spine (Phila Pa 1976) 2018, 43(20); Sci Rep 2020, 10(1), 4115; Front Bioeng Biotechnol 2022, 9, 807679 ). Furthermore, all tests and experiments were performed in triplicate, both technical and experimental. After verification of normal distribution and homogeneity of variance through theLevene test, our data were assessed with t-test to compare female and male MSCs from vBMAs clot.
- In Figure1 and Figure5, please add the unit of longitudinal axis in the graph.
We added the unit of longitudinal axis.
- In Table1, the proportions of MSCs with different phenotype were presented. Please provide the original flow chart of flow cytometry.
As suggested, we provided the flow chart of flow cytometry.
- In Figure6, representative images of osteogenic, adipogenic, and chondrogenic differentiation ability were presented. Please add the Western Blot of remarkable hallmarks with regard to the pathways in the process of osteogenic, adipogenic, and chondrogenic differentiation.
Although this aspect would be very interesting, we did not and no longer have the necessary amount of biological material (vBMA clot and/or derived MSCs) to perform Western Blot of remarkable hallmarks with regard to the pathways in the process of osteogenic, adipogenic, and chondrogenic differentiation.
- In Figure 7, except for Klotho gene, the senescence-associated genes listed are associated with inflammation. Are there other specific genes related to senescence?
Cellular senescence can be identified as a loss of proliferative capacity after extended culture, which is known as replicative senescence. The colony-forming unit-fibroblast (CFU-F) assay is one of the most frequently used qualitative methods to estimate and evaluate MSCs’ proliferation potential in vitro. Senescent MSCs exhibit a decreased level of CFU-F, and, more importantly, they display smaller colony sizes (Exp Cell Res. 2004;294(1):1-8). Additionally, it was reported that in senescence, MSCs display specific secretions that regulate and maintain the aging phenotype (Stem Cells Transl Med. 2022 Apr; 11(4): 356–371). These secretions include IL1β, IL1α, IL6, IL8, CCL4, CXCL2, TNFα, and MCP-1, and a variety of secretory cytokines and chemokines. It is proven that SASP enables senescent cells to participate in remodeling their environment through modulation of multiple physiological functions including wound healing and embryonic development. In addition, secretory cytokines, growth factors, and proteinases of senescent MSCs are reported to be not only aging and inflammatory markers but also aging triggers in senescence of MSCs derived from human tissues. Indeed, SASP from MSCs of bone marrow and adipose tissues were analyzed and reported with signaling ability to maintain and induce senescence in their niche (Aging. 2016;8(7):1316–29). Obviously, senescent cells produce and secrete other factors, such as IL-3, IL-4 and IL-17, epidermal growth factor (EGF), fibroblast growth factors-2 (FGF-2), FGF-4 and FGF-8, insulin growth factor-1 (IGF-1), which are also referred as SASP factors.
- Except for the MSCs many other cellular components co-exist in the marrow microenvironment. Whether the cross-talk between MSCs and other cells exists in the bone regeneration?
As supposed by the reviewer a crosstalk between MSCs and other cells and growth factors exists during bone regeneration. In fact, bone marrow aspiration is an easy, safe and inexpensive method that makes possible a direct transplantation MSCs, endothelial progenitor cells, hematopoietic stem cells, other progenitor cells, growth factors, e.g. bone morphogenetic proteins (BMPs), platelet-derived growth factor (PDGF), transforming growth factor-β (TGF-β), vascular endothelial growth factor (VEGF), and several interleukins, all of them able to stimulated bone healing and regeneration. Despite only a very small percentage (0.01–0.001%) of MSCs is found among the totality of mononuclear cells in BMA, the attendance of non-adherent osteogenic cells and the potential collaboration among BMA cell types in tissue repair suggest that the use of whole BMA, instead of expanded, concentrated and purified MSCs, can be a better bone cell therapy approach.
Round 2
Reviewer 1 Report
Salamanna and coauthors made significant efforts to address the issues raised in the initial review comments and greatly improved the overall quality of the revised manuscript. The language is much easier to read now, and many issues related to figures and results were satisfactorily addressed. However, there are still several obvious issues that should be addressed before publication.
1. For Figure 1, the title on line 100 “Enzyme-linked immunosorbent assay (ELISA) growth factors expression” is not correct grammatically. Also, the y-axis title should include “(ng/ml)” to match the values for proper interpretation. With “(ng/ml)” labeled in y-axis, there will be no need to add “(pg/ml)” and “(ng/ml)” after the protein names in the legend. The “(ng/ml)” after PDGF-C is an obvious mistake. The same issue applies to lines 367-378.
2. For Figure 2, the red fluorescence should be noticeable after the green and red overlay. The authors indeed showed the red single channel images in the response to the 3rd comment of the initial review report, but that is not enough to address the issue. Since the authors counted the dead cells, then the dead cells (the red fluorescence) should be represented in the overlaid images in Figure 2. It may worthwhile to increase the brightness and contrast of the red channel to make it noticeable.
3. In line 131, remove the “,” before “of CD73”.
4. “2−ΔCt” in lines 208, 221 and 443 should be “2−ΔCt”. Question: should it be “2−ΔCt” or “2−ΔΔCt”?
5. For the comment #11 in the initial review report, the issue about the inaccurate expressions of the cell numbers still exists. For example, the “a density of 7 × 103 cells per cm2” in line 411 should be “a density of 7 × 103 cells per cm2”. The incorrect expression of the numbers is prone to misunderstanding.
The language is greatly improved and much easier to read now
Author Response
- For Figure 1, the title on line 100 “Enzyme-linked immunosorbent assay (ELISA) growth factors expression” is not correct grammatically. Also, the y-axis title should include “(ng/ml)” to match the values for proper interpretation. With “(ng/ml)” labeled in y-axis, there will be no need to add “(pg/ml)” and “(ng/ml)” after the protein names in the legend. The “(ng/ml)” after PDGF-C is an obvious mistake. The same issue applies to lines 367-378.
As suggested by the reviewer we correct the Figure 1 title, figure legend and lines 367-378.
- For Figure 2, the red fluorescence should be noticeable after the green and red overlay. The authors indeed showed the red single channel images in the response to the 3rd comment of the initial review report, but that is not enough to address the issue. Since the authors counted the dead cells, then the dead cells (the red fluorescence) should be represented in the overlaid images in Figure 2. It may worthwhile to increase the brightness and contrast of the red channel to make it noticeable.
As suggested, we increased the brightness and contrast of the red channel to make it noticeable in Figure 2.
- In line 131, remove the “,” before “of CD73”.
We corrected as suggested.
- “2−ΔCt” in lines 208, 221 and 443 should be “2−ΔCt”. Question: should it be “2−ΔCt” or “2−ΔΔCt”?
We corrected as request. We used 2−ΔCt and not 2−ΔΔCt.
- For the comment #11 in the initial review report, the issue about the inaccurate expressions of the cell numbers still exists. For example, the “a density of 7 × 103 cells per cm2” in line 411 should be “a density of 7 × 103 cells per cm2”. The incorrect expression of the numbers is prone to misunderstanding.
As suggested by the reviewer we corrected the expression of the cell’s number.
Reviewer 2 Report
Thanks for the revision and detailed response. This revised version was suitable for this journal.
Author Response
Thanks for the revision and detailed response. This revised version was suitable for this journal.
We thank the reviewer for the comment.